# How Periodontal Disease and Presence of Nitric Oxide Reducing Oral Bacteria Can Affect Blood Pressure

**DOI:** 10.3390/ijms21207538

**Published:** 2020-10-13

**Authors:** Pamela Pignatelli, Giulia Fabietti, Annalisa Ricci, Adriano Piattelli, Maria Cristina Curia

**Affiliations:** Department of Medical, Oral and Biotechnological Sciences, “G. d’Annunzio” University of Chieti-Pescara, Via dei Vestini, 66100 Chieti, Italy; pamelapignatelli89p@gmail.com (P.P.); giuliafabietti@gmail.com (G.F.); annalisa.ricci95@gmail.com (A.R.); apiattelli@unich.it (A.P.)

**Keywords:** nitric oxide, oral reducing bacteria, endothelial dysfunction, blood pressure, hypertension, mouthwash

## Abstract

Nitric oxide (NO), a small gaseous and multifunctional signaling molecule, is involved in the maintenance of metabolic and cardiovascular homeostasis. It is endogenously produced in the vascular endothelium by specific enzymes known as NO synthases (NOSs). Subsequently, NO is readily oxidized to nitrite and nitrate. Nitrite is also derived from exogenous inorganic nitrate (NO_3_) contained in meat, vegetables, and drinking water, resulting in greater plasma NO_2_ concentration and major reduction in systemic blood pressure (BP). The recycling process of nitrate and nitrite to NO (nitrate-nitrite-NO pathway), known as the enterosalivary cycle of nitrate, is dependent upon oral commensal nitrate-reducing bacteria of the dorsal tongue. *Veillonella*, *Actinomyces*, *Haemophilus*, and *Neisseria* are the most copious among the nitrate-reducing bacteria. The use of chlorhexidine mouthwashes and tongue cleaning can mitigate the bacterial nitrate-related BP lowering effects. Imbalances in the oral reducing microbiota have been associated with a decrease of NO, promoting endothelial dysfunction, and increased cardiovascular risk. Although there is a relationship between periodontitis and hypertension (HT), the correlation between nitrate-reducing bacteria and HT has been poorly studied. Restoring the oral flora and NO activity by probiotics may be considered a potential therapeutic strategy to treat HT.

## 1. Introduction

There is a known correlation between oral health and systemic disease [1]. Particularly significant evidences associate periodontal bacteria and tooth loss to systemic disorders and specifically to cardiovascular disease, such as high BP. Furthermore, a correlation between periodontal disease and hypertension has been recently reported [2].

Through nitrate-nitrite reduction, some commensal oral bacteria can supply bioactive NO, essential for the endothelial cell function and regulation of arterial BP [3,4]. Consequently, it is thought that a decreased quantity of oral nitrate-reducing bacteria and an increased quantity of pathogenic bacteria, are responsible for a correlation between oral hygiene and chronic periodontitis and, at a later stage, cardiovascular diseases (CVD).

NO is a free radical and simple gas that is synthesized endogenously by a family of enzymes namely NOSs [5]. Normally, NO is produced from the amino acid L-arginine in the presence of oxygen by eNOS and it has an important role in preserving vascular homeostasis. NO is a multifunctional signaling molecule involved in the maintenance of metabolic and cardiovascular homeostasis and also a potent endogenous vasodilator that suppresses the formation of vascular lesions in atherosclerosis.

Imbalance in NO bioavailability is associated with some cardiovascular and metabolic diseases [6]. Reduction of oxygen provision, such as in the case of myocardial ischaemia, compromises NO synthesis. In addition to being synthesized by NOS, nitrogen can derive from the bioconversion of nitrates, originating from the diet and in particularly from vegetables and drinking water, into nitrite by means of oral bacteria, through a process known as NO_3_-NO_2_-NO reduction pathway [7]. This pathway is capable of reducing BP in either healthy young or old adults.

Decreased production or activity of NO, due to endothelial dysfunction, is responsible for the pathogenesis of many cardiovascular diseases, including atherosclerosis and CVD such as hypertension, coronary artery disease [8,9,10,11,12].

The aim of this review is to investigate the correlation between the oral nitrate-reducing bacteria and measured BP in healthy and hypertensive individuals.

## 2. NO_3_-NO_2_-NO Reduction Pathway

### 2.1. Endogenous NO and NOS

NO homeostasis is crucial to mammalian physiology. The discovery of endothelium derived relaxing factor (EDRF) in 1980 [13] that was later recognized as NO [14] revolutionized vascular biology and identified new strategies to control BP. NO is a small and lipophilic molecule produced by specific NOSs enzymes, that use oxygen and NADPH to oxidize L-arginine to L-citrulline. Endogenous NO is produced by endothelial cells and acts as a vasodilator [13,14]. The NOS enzymes oxidize L-arginine by using haem, tetrahydrobiopterin, and flavin reductases as cofactors. The scarcity of molecular oxygen during hypoxia makes these enzymes somewhat malfunctioning [15]. Hemoglobin and myoglobin are hemeproteins that convert nitrite into NO because they perform nitrite reductase activity when they are oxygenated. On the contrary, deoxygenated hemoglobin and myoglobin oxidize NO and nitrite to nitrate [16,17].

Three NOS isoforms have been studied and they take their names from the tissues they identified: Neuronal NOS (nNOS, NOS 1), inducible NOS (iNOS, NOS 2), and endothelial NOS (eNOS, NOS 3) [18]. The nNOS was discovered in the brain and is involved in central and peripheral neuronal signal [19]. The iNOS, the inducible form of NOS, detected in macrophages, produces high levels of NO when such cells are activated. These high levels of NO can become toxic towards bacteria, virus, and tumor cells [7]. The expression of iNOS is stimulated by various cytokines [20]. The eNOS has been identified in endothelial cells and regulates vascular tone and endothelial integrity [21,22]. NO moves from the vessels, where it is produced, to the surrounding muscle cells, favoring their relaxation. This process generates vasodilation and increases blood flow. It also produces a replenishment of oxygen and ultimately a reduction in BP.

Immediately after its production NO is oxidized to nitrite and nitrate [23,24] and the effectiveness of the production is measured by the presence of anions [24,25]. The recycling of nitrate and nitrite from the production of NO is known as the enterosalivary cycle of nitrate [26]. Nitrite is reduced to NO and other reactive nitrogen intermediates via nitration (-NO_2_), direct NO signaling, and nitrosation (-NO). Among NO functions, vasodilation is possible through the activation of guanylyl cyclase leading to cGMP production [27]. NOS forms S-nitrosothiols; NO reduced from S-nitrosothiols and NO_2_^−^ can exert classical cytotoxic and cyclic GMP (cGMP)-dependent effects, such as vascular smooth muscle relaxation. S-nitrosothiols are vasoactive endogenous compounds formed in the stomach as a result of nitrite achievement. S-nitrosylation is probably involved in the lowering of blood pressure by both nitrite and nitrate [28].

### 2.2. Dietary Nitrate

In addition to the oxidation of NO, nitrite is also derived from meat, vegetables as beetroot, lettuce, spinach, and drinking water that represent high natural dietary sources of inorganic nitrate (NO_3_). After being ingested, nitrate returns to the oral cavity through the blood circulation and reaches the salivary glands; finally, it is reduced to nitrite by the oral microflora [27,28] (Figure 1).

Qin et al. (2012) identified sialin as an effective nitrate transporter and as the major inorganic nitrate uptake system in salivary gland cells. From here, the nitrate is secreted in the oral cavity where commensal anaerobic bacteria particularly abundant in the dorsal tongue reduce it to nitrite using the nitrate reductase enzymes [3,29]. Therefore, the salivary nitrite produced is then ingested and subsequently transformed, in the acidic gastric environment, first into nitrous acid and then in NO and other bioactive nitrogen oxides [30,31] but also in N-nitrosamines, a class of carcinogen. However, some data suggest a protective role of salivary nitrite in the stomach [32]. Furthermore, the nitrogen intermediates, generated by high concentrations of nitrate and nitrite in normal saliva, are involved in host defence, having a potential protective effect including an anti-bacterial function. Therefore, when salivary nitrite reaches the acidic gastric environment and turns into nitrogen species, it provides beneficial effects not only in the stomach but also systemically, participating in the first-line defence against aggressors and bacteria [32]. Dietary nitrate improves NO bioavailability, that is the production and utilization of endothelial NO in organisms, and increases vasodilatation. About 24 h after the consumption of nitrate-rich food, NO is absorbed in the upper gastrointestinal tract and three-quarters of the nitrate is excreted in the urine [3,33]. Moreover, faecal microbiota can reduce NO_3_^−^ to NO via reduction to ammonia and despite the fact that NO has a short half-life and immediately oxidizes into nitrates and nitrites, it can be stored in the form of S-nitrosothiols in red blood cells and in plasma, where it maintains a physiological effect and represents a marker of NO availability [34].

In addition to hemeproteins, dietary nitrite absorbed in blood and tissues can also be reduced by other proteins such as mitochondrial enzymes. In fact, the role of mitochondria in the nitrite reduction has been reported by several authors [35,36,37,38].

In addition to being a key enzyme of the purine degradation pathway, Xanthine oxidoreductase (XOR) reduces oxygen to superoxide and hydrogen peroxide and its last action can contribute to the nitroso-redox alteration observed in hypertension [39,40]. Furthermore, XOR can also reduce nitrate to nitrite and then to NO, thus acting as a protective mechanism [41,42].

## 3. The Role of Oral Microbiome in NO Homeostasis

Oral cavity commensal bacteria can use nitrate and reduce it into nitrite, which can become vasoactive and work as a potent vasodilator. The nitrate-nitrite-NO pathway is affected by microbial diversity [43] and oral pathology [44]. Some microbes live in symbiosis with their host, and this can significantly contribute to health [45,46]. Human microbiome is capable of exerting effects on their host through the production of metabolites, immune responses, and gene expression. Dysbiotic states due to low diversity of microbial species may promote chronic inflammatory status related to obesity, diabetes, metabolic syndrome, adverse events in pregnancy and cardiovascular disease [47]. There is a symbiotic relationship between oral nitrate-reducing bacteria and humans in which bioactive NO was produced from dietary nitrate and nitrite reduction. This has been described as enterosalivary nitrate-nitrite-NO pathway and it contributes to host NO homeostasis. Conversely, the dysbiosis of nitrate-reducing bacteria can affect the nitrate-nitrite-NO pathway, causing NO impairment. The nitrate is first transported to the trans-membrane; subsequently, the organism evaluates the quantity of nitrite that can become cytotoxic and at this point reduces the bacteria act. The presence of reducing bacteria affects nitrite and NO bioavailability indirectly by decreasing the risk of cardiovascular disease. Oral nitrate-reducing bacteria contribute to circulating NO and thus the dilation capacity of arteries [4,48].

### 3.1. Oral Nitrate-Nitrite Reducing Bacteria

A key role has been attributed to mammalian entero salivary commensal bacteria. The most abundant nitrate-reducing species localized in crypts of the tongue emerge to be: *Staphylococcus*, *Streptococcus*, *Actinomyces*, *P. melaninogenica*, *V. dispar*, *H. parainfluenzae*, *N. subflava*, *V. parvula*, *F. nucleatum* subsp. *nucleatum*, *C. concisus*, *L. buccalis*, *P. intermedia* [49]. This area of the mouth has been previously shown to contain a high abundance of NO_3_^−^ reducing Gram-negative bacteria as they favor the anaerobic environment provided by the deep crypts of the tongue [50]. The posterior papillary structure can accumulate a larger amount of biofilm than other areas of oral mucosa [51]. The bacteria present on the posterior dorsal tongue are different from those proliferating on teeth and in the gingival pocket. They are much more similar to saliva microbiome [52]. Oral nitrate is converted into nitrite and NO by *Actinomyces*, *Haemophilus*, *Neisseria,* and *Veillonella*. Then, NO is absorbed through the epithelial vascularization or through the gastrointestinal processing [43]. The concentration of oral bacteria changes depending on the individual variability and the method of analysis used [52]. Due to culturing techniques on solid media it was possible to begin the study of oral bacteria [53,54]. Then, the introduction of nucleic acid analysis techniques, DNA hybridization, Polymerase Chain Reaction, and Sanger 16S rRNA sequencing, combined recently with high-throughput pyro-sequencing-based analyses and whole genome shotgun sequencing (WGS) allows more accurate taxonomic assignment at the species level and the most complete qualitative and quantitative knowledge of human oral microbiome [55].

Koopman et al. also demonstrated a loss of nitrate-reducing bacteria community diversity when saliva samples were cultured before sequencing. This may have resulted in under-estimating the results. Incubated biofilm samples were dominated by *Streptococcus*, in contrast with what was observed after WGS sequencing. These data revealed that communities of oral bacteria change in culture (some grow and others do not) [56]. The use of 16S rRNA gene pyrosequencing enabled the identification of 14 candidate species that were present at an abundance of at least 0.1% in communities with the best nitrate reduction [4,43]. Among tongue scraping nitrate-reducing bacteria the most abundant were *Veillonella* species, strict anaerobes, *Prevotella*, *Neisseria*, *Heemophilus,* and *Actinomyces* spp. [4].

By using the principal coordinate analysis (PCoA), samples with high nitrate reduction capacity were more likely to contain also *Granulicatella adiacens*, *Actinomyces odontolyticus*, *Fusobacterium nucleatum*, *Veillonella dispar*, and some unclassified species of the Gemellaceae family [57]. Some of these (*Actinomyces odontolyticus*, *Fusobacterium nucleatum*, and *Veillonella dispar*) are associated with poor oral health. Low diversity of microbial species results in dysbiotic states, which can compromise the nitrogen cycle, exacerbating pre-existing microbiota dysbiosis.

### 3.2. Interaction among Oral Microbiota, Dietary Nitrate, and NO Homeostasis

The production of diet nitrites is influenced by gastrointestinal microflora, mammalian cells in the gut wall, and nitrate-reducing oral bacteria. The microbiota is capable of producing NO chemically, biologically, and via the NOS pathway [58]. Nitrate-reducing oral bacteria are responsible for the bioactivation of exogenous nitrate, derived from green vegetables and endogenous nitrate, transforming it into nitrite and NO [50]. Only 25% of exogenous nitrate is absorbed by the salivary glands and concentrated up to 20-fold in saliva (1500 µM) [59,60].

The abundance of NO_3_ reducing bacteria was significantly correlated with the change in NO_2_ salivary (P = 0.03, r = 0.44) [61]. It has been suggested that bacteria do not work only independently but in synergy. Indeed, *A. odontolyticus* possesses only nitrate reductase encoding genes in their genomes. In contrast, *S. mutans* and *F. nucleatum* do not reduce nitrate but nitrite. *F. nucleatum* acts as a driver for the attack of later microbes and favors biofilm formation and proliferation of main reducing bacteria during dietary intake [62]. There is a bacterial association contributing to nitrate reduction and nitrite accumulation. NO_3_ supplementation increases plasma concentration of NO_2_^−^ and reduces systemic BP in old (70–79 years), but not young (18–22 years) participants, as it was reported by Vanhatalo et al. [62]. At the same time, in response to NO_3_ supplementation, high abundances of *Rothia* and *Neisseria* and low abundances of *Prevotella* and *Veillonella* are correlated with greater increases in plasma [NO_2_^−^].

Velmurugan et al. established that 6 weeks of NO_3_ supplementation with nitrate-rich beetroot juice (BR) resulted in improved vascular function, a significant increase in the percentage of *Neisseria flavscens* in the oral microbioma and an increase of *Rothia mucilaginosa*, known as NO_3_ reduction [63]. This study has shown that the oral microbiome may be altered by increased dietary NO_3_ provision. This leads to an improvement in individual capacity to reduce ingested NO_3_ and furthermore in greater plasma NO_2_ concentration and BP reduction. The increase of plasma [NO_3_^−^] and [NO_2_^−^] is correlated with high abundances of *Rothia* and *Neisseria* and low abundances of *Prevotella* and *Veillonella* in oral cavity. A high relative concentration of *Prevotella melaninogenica* was associated with a greater score in Salivary Flow Rate Questionnaires (SFR-Q) and smaller changes in plasma [NO_2_^−^], systolic blood pressure (SBP), and pulse-wave velocity (PWV) in response to BR supplementation. Saliva samples represent a compound of bacteria from all oral sites, while dorsal tongue swab samples show the highest nitrate reductase activity [62,64].

PWV is positively correlated with the relative abundance of *Rothia* and *R. mucilaginosa* which may promote vascular health.

Dietary NO_3_ supplementation may encourage an increase in *Rothia* and *Neisseria* but at the same time it reduces the concentration of Prevotella and Veillonella. Indeed, the ratio of Neisseria to Prevotella is significantly higher in vegans compared to omnivores [65]. The abundance of NO_3_ reducing bacteria on the dorsal surface of the tongue significantly correlates with salivary NO_2_ generation following diet supplementation.

After NO_3_ supplementation, *F. nucleatum* subspecies and *Actinomycetales*, obligate anaerobes such as *Actinomyces odontolyticus* and *Actinomyces naeslundii*, were associated with greater increase in plasma [NO_2_-] and greater reductions in BP, while *C. concisus* and *P. melaninogenica*, particularly over-represented in tongue swab samples at baseline, were associated with consumed NO_2_ [66].

However, a higher quantity of oral reducing bacteria does not seem to increase plasmatic nitrite levels after ingestion of beetroot juice.

## 4. Effects of Antiseptic Mouthwash on Oral Microbiota and BP

Mouthwashes can affect the oral concentration of reducing bacteria and thus influence the NO_3_ metabolism. The reported data are discordant. In fact, four studies [67,68,69,70] have described a significant negative action of mouthwashing on plasma nitrite concentration (*p* < 0.05), meanwhile two other studies [67,71] have found no significant alteration of plasma nitrate and nitrite, by comparing mouthwash administration against placebo. Use of 0.2% chlorhexidine mouthwash twice a day for 7 days reduced salivary NO release by almost 50%, 90 min after nitrate loading and increased systolic and diastolic BP by 2–3.5 mmHg [68]. The effect of chlorhexidine mouthwash on resting SBP is comparable to changes of at least 5 mm/Hg induced by manipulation of dietary salt intake [72]. Woessner et al. reported inhibition in the production of salivary nitrite after the ingestion of 140 mL of beetroot juice (8.4 mM or 521 mg of nitrate) both after the use of mouthwashing with essential oil and of cetylpyridinium chloride 0.05% [70]. The daily use of chlorhexidine mouthwash, even at a low concentration of 0.0025%, may reduce the count of *Veillonella dispar* and inhibit nitrate-reducing activity but it does not affect salivary nitrite production. Essential oil and povidone-iodine mouthwash has proven to have little effect on nitrate-reducing activity [73,74]. On the contrary, Senkus et al. reported that 0.2% chlorhexidine mouthwash twice a day resulted in a significant reduction in salivary and plasma nitrite with a concomitant rise in salivary and plasma nitrate concentrations. Clinical manifestation of these biochemical changes was a significant increase in SBP (2.3 mmHg increase, *p* = 0.01). BP and plasma nitrite concentrations were inversely related. [75]. It was noticed that there was an inter-subject variability in response to 7 days of treatment with chlorhexidine dependent upon the composition of the baseline tongue microbiota with a significant change in SBP between the treatment and 3-days recovery time points (115 vs. 111.5 mmHg) [76]. Tongue bacterial community composition was influenced by daily cleaning (either once or twice) [43]. Indeed, regular tongue cleaning can be responsible for an increased proportion of *H. parainfluenza* and other Proteobacteria, major nitrate and nitrite reducing species [57]. Furthermore, the cleaning of the tongue may disrupt the papillary surface and cause an increased penetration of chlorhexidine, which may result in a significant disruption of community alpha-diversity and consequently in a greater SBP. Rapid nitrate/nitrite reducing bacteria recovery is observed in patients with frequent tongue cleaning as well as an increase in the nitrate reduction metabolic activity. This may also activate bacterial metabolism [76]. Studies on animal models suggest that the use of chlorhexidine mouthwash prevents nitrite formation in the oral cavity, consequently nitrite does not enter the acidic environment of the stomach thus weakening the s-nitrosylation process [28].

## 5. Endothelial Disfunction, NO Bioavailability, and BP

The endothelium controls the blood vessel tone by influencing the release of vasoactive mediators including NO. The endothelial-derived NO production is important for the maintenance of a healthy endothelium [26,77] and in turn the integrity of the endothelium is crucial for the maintenance of vascular homeostasis and for a healthy cardiovascular system. In healthy individuals, the production of eNOS causes vasodilation in either muscular and resistance vessels. In contrast, in patients with atherosclerosis or endothelial dysfunction, the low bioavailability of NO causes an attenuated vasodilation in peripheral vessels and a paradoxical vasoconstriction in coronary arteries [78,79].

Emerging evidence shows that endothelial dysfunction can decrease circulating levels of nitrite [80,81] and subsequent NO deficiency is critically associated with the development of hypertension and other forms of vascular and cardiovascular disease [82]. Nitrate and/or nitrite oral supplementation promotes the increase of circulating nitrite levels and prevents endothelial dysfunction in mice [83,84], healthy adults [85,86,87,88], obese individuals [89], and hypercolesterolemic patients [63]. Oxidative stress, lipid infiltration, expressions of some inflammatory factors, and alteration of vascular tone play an important role in endothelial dysfunction by causing a decrease in NO bioavailability.

Imbalances in the oral reducing microbial community have been associated with reduced cardiovascular and metabolic health.

Burleigh et al. [61] found that the abundance of nitrate-reducing bacteria was associated with the generation of salivary nitrite. Furthermore, a higher quantity of reducing bacteria is not necessarily related to higher NO bioavailability, measurable with plasma nitrite response, and does not affect the generation of NO through the diet. This is because there are individuals with compromised NO bioavailability, including older adults, patients with endothelial dysfunction, and those treated with antibiotics. Among nitrate-reducing bacteria of the mouth, Prevotella is the most abundant. According to the previous work [57,61,66], Prevotella and Veillonella are the first and second most copious genera among known NO_3_ reducing bacteria, contrasting with previous research that identified only *Veillonella* as the most abundant taxa found on the tongue dorsum [57]. *Veillonella*, *Prevotella*, *Haemophilus*, and *Streptococcus* were found in abundance on the tongue dorsum of healthy individuals [52]. Through 16S RNA sequencing, only seven of fourteen known species which have previously been demonstrated by Hyde et al. to reduce NO_3_ in vitro were detected. In the work by Hyde et al. [57], the analysis by “shotgun” WGS identified bacterial species of three tongue scrape samples. Given that WGS sequences all genes rather than the more targeted approach of 16s RNA sequencing, this method explains the differences between the results obtained with the two methods.

## 6. Periodontitis Affects Hypertension (HT)

Hypertension is a multifactorial disease associated with impaired NO production and bioavailability, affecting 34% of adults 20 years and older and 67% of adults 60 years and older in the United States [90]. It is known that there is an interplay between periodontal disease and HT, as demonstrated by the greater presence of periodontitis in CVD patients with HT than in those without HT. Furthermore, specific periodontopathic bacteria *A. actinomycetemconcomitans* and *P. intermedia* were highly detected in male HT subjects compared to non-HT subjects, suggesting an increased risk in individuals with untreated chronic periodontitis. This indicated a relationship between HT and specific periodontopathic bacterial infection [91].

As it is known that HT is an important risk factor for stroke [92] and, as periodontal infections contribute to the development of hypertension, we would therefore expect them to be greatly associated with stroke. There is also a correlation between periodontal disease and either subclinical atherosclerosis [93,94,95] and endothelial dysfunction [96]. Desvarieux et al. [97] also reported a strong positive association between increased subgingival colonization by *A. actinomycetemconcomitan*, *P. gingivalis*, *T. forsythia,* and *T. denticola* and prevalent hypertension.

Periodontitis has effects on the endothelial function, causing endothelial dysfunction, stiffness of the arteries and high BP [98], and hypertension-associated oral pathogens (HOP) as *C. rectus*, *V. parvula*, *P. melaninogenica* have been identified [99]. Late colonizers as *P. gingivalis*, *A. actinomycetemconcomitan*, *T. forsythia,* and *T. denticola* have been explored in vascular atherosclerosis lesions but their possible association with blood pressure has not been evaluated [100,101]. Hypertension, such as periodontitis, is considered a low-grade inflammatory disorder and it is marked by the presence of pro-inflammatory cytokines [102]. These bacteria are inducers of potent pro-inflammatory cytokines, as interleukin (IL)-1b, 6, 8, and tumor necrosis factor (TNF) alfa. Furthermore, inflammatory mediators as IL-6 have been associated with decreased NO production, a key factor in the development of high BP [103].

Few studies reported the correlation between subgingival bacteria and blood pressure. Gordon et al. [104,105] reported a less abundance of subgingival *Prevotella* oral and *Streptococcus oralis* in postmenopausal women treated with antihypertensives compared to non-treated women. Both bacteria are periodontal pathogens and are associated with vascular disease. Pathogenic bacteria as *P. gingivalis*, *T. forsythia*, *T. denticola*, and *F. nucleatum* were positively correlated with mean SBP and DBP but none of them was known as nitrate-reducing bacteria.

## 7. Conclusions

Organic nitrate reduction to nitrite is catalyzed by the nitrate reductase enzyme produced by commensal anaerobic microbiota particularly abundant on the posterior dorsal surface of the tongue. The nitrite is then converted to NO in the acidic gastric environment and in the systemic circulation by eNOS. The “excess” of NO_3_ and NO_2_ from saliva, coming from both exogenous and endogenous origins, is excreted in the urine, thus preventing an excessive BP drop. Some mitochondrial enzymes exert a protective effect in the nitrite reduction particularly in conditions as ischemia and reperfusion.

Nitrite supplied by the solution or beetroot juice has been reported to lower SBP. Dietary nitrate and its sequential reduction to nitrite and NO in the circulation are responsible for the cardioprotective effects of reducing BP with the promotion of vasodilatation, increasing endothelial functions, promoting the release of circulating angiogenic cells from the bone marrow, and inhibiting platelet aggregation.

Nitrate-reducing oral bacteria have a direct role in the enterosalivary pathway of NO production and mediate the BP effect of dietary nitrate, boosting oral and plasmatic NO_3_ bioavailability. Nitrate/nitrite-reducing bacteria are cooperating synergistically creating an optimal oral bacterial community for NO generation. They are protective on cardiometabolic health contrary to periodontal pathogenic bacteria associated with an increased risk of adverse cardiovascular events. The daily use of mouthwashes with chlorhexidine, essential oil, cetylpyridinium chloride, and routine tongue brushing can affect oral concentrations of reducing nitric oxide bacteria.

In particular, the chlorhexidine antiseptic mouthwash prevents the reduction of nitrate to nitrite by oral commensal bacteria, disrupts the enterosalivary cycle of nitrate, and abolishes hypotensive effects of nitrate. The chlorhexidine antiseptic mouthwash may increase oral microbiome diversity disrupting some species and allowing the proliferation of others. It remains to be clarified whether the use of antiseptic mouthwash disrupts the enterosalivary cycle as a result of decreased or altered oral microbiome.

The prospective to restore the oral microbiome by probiotics to increase NO bioavailability represents a new strategy in cardiovascular medicine and dentistry. Therefore, providing NO generation by using nitrite and nitrate may be considered a potential therapeutic approach to the management of resistant hypertensive patients.

## Figures and Tables

**Figure 1 ijms-21-07538-f001:**
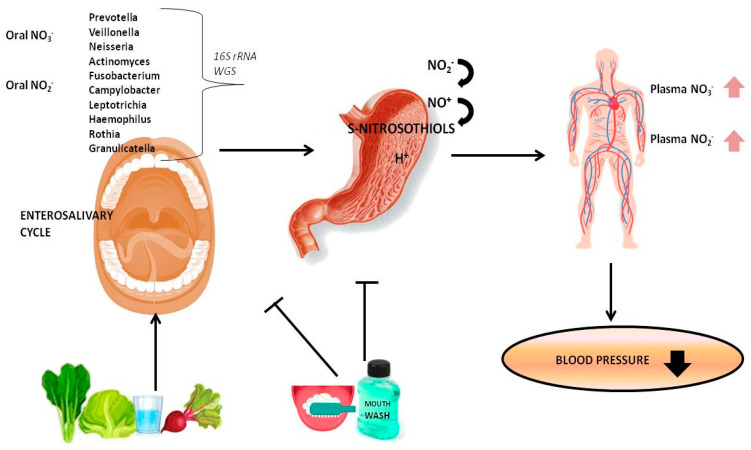
Interplay between dietary nitrate and oral commensal bacteria in the enterosalivary circulation of nitrate and in blood pressure. In the figure, the most reported abundant nitrate-reducing bacteria are listed, detected both by Sanger 16S rRNA sequencing and by whole genome sequencing (WGS). By the shotgun WGS, a method that sequences all genes rather than the more targeted approach of 16S rRNA sequencing, a greater number of bacterial species have been identified (see main text). The increase in nitrates and nitrites in the plasma has the effect of dilating the capacity of the arteries and consequently lowering blood pressure. Dietary nitrate and oral reducing bacteria contribute to nitrate reduction and circulating nitrous oxide. On the contrary, the daily use of an antiseptic mouthwash and tongue cleaning prevents the reduction of nitrate to nitrite and disrupts the enterosalivary cycle by affecting oral concentrations of commensal reducing bacteria. Therefore, nitrite does not achieve the acidic environment of the gastric cavity thus decreasing the S-nitrosothiols formation.

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
