# Peer review of "How Periodontal Disease and Presence of Nitric Oxide Reducing Oral Bacteria Can Affect Blood Pressure"

_ijms, 2020, doi:10.3390/ijms21207538_

Round 1

Reviewer 1 Report

P. 3 line 103; 104: Also faecal microbiota can reduce NO3  to NO via reduction to ammonia and, despite…

please clarify the sentence
p. 3 line 104; p.6 lines 265; 268: Please explain why the text is
underlined? What does it mean?

Author Response

We thank you for reviewing the manuscript and making comments and suggestions.

P.3 line 103: please clarify the sentence

P.3 line 104:please explain why the text is underlined.

The sentence break and the underlining are simply typographical errors. We have modified the sentence.

Reviewer 2 Report

This manuscript reviews the literature on NO reduction pathways, possible roles of oral microbiome in NO homeostasis, and possible correlation of presence of nitric oxide reducing oral bacteria with blood pressure.

The manuscript is not well prepared, with many spelling and grammatical errors, and even the essential chemical terms/formulas were used wrong. With only one Figure, the whole manuscript is mere text with poor illustration, not appearing informative. The Objectives and Conclusion are not well-supported by the shallow “touch and go” accounting of the relevant literature.

In conclusion, major revision is necessary before further consideration.

Some details:

1. The manuscript is not well prepared, with many spelling and grammatical errors. From careful reading of the first 2 pages this reviewer has found the following errors, nealy every paragraph has some. Careful proofreading by a native English speaker is necessary.

L21, should read “has been poorly studied.”

L30, “such as”

L30, the last statment needs references.

L33, The sentence “For this reason…” needs to be rewritten.

L47, The sentence “The pathway…” needs to be rewritten.

L49, should read “is responsible for”Instead the association between

297 reducing bacteria and hypertension has never been studied so far.

L53, is it BP or hypertension status? Which one?

L58, “identified” is not a proper word.

L61, check spelling.

L62, “more or less” should not appear in an scientific article.

L64, Sentence “Instead…” needs to be rewritten.

L68, Sentence “ nNOS...[6]” needs to be rewritten.

L81, “NOS...(GC)” needs to be rewritten.

2. It does not appear that the authors understand the fundamental chemistry: even the essential chemical formulas are all written wrong!

In NO2 and NO3, 2 and 3 should be subscripted.

NO3- if the negative sign is for negative charge, it needs to be superscripted.

Fig. 1 same problem: NO-3, 3 needs to be put right after O, NO-2, 2 needs to be put right after O.

2. The manuscript is mere literature report, with minimal effort of dissemination of useful information into strong arguments supporting the objectives the authors have set out to do, demonstrating “How periodontal disease and presence of nitric oxide reducing oral bacteria can affect blood pressure.” Hence, the correlation is not immediately available and the Conclusion remains unsupported. And with statements, such as “Instead the association between reducing bacteria and hypertension has never been studied so far,” the authors failed to present a strong case.

Unfortunately, the few published studies that support the Conclusion, were given very little account. More details should be given.

Author Response

We thank you for reviewing the manuscriptand making comments and suggestions.

Comments and Suggestions for Authors

This manuscript reviews the literature on NO reduction pathways, possible roles of oral microbiome in NO homeostasis, and possible correlation of presence of nitric oxide reducing oral bacteria with blood pressure.

The manuscript is not well prepared, with many spelling and grammatical errors, and even the essential chemical terms/formulas were used wrong. With only one Figure, the whole manuscript is mere text with poor illustration, not appearing informative. The Objectives and Conclusion are not well-supported by the shallow “touch and go” accounting of the relevant literature.

In conclusion, major revision is necessary before further consideration.

Some details:

The manuscript is not well prepared, with many spelling and grammatical errors. From careful reading of the first 2 pages this reviewer has found the following errors, nearly every paragraph has some. Careful proofreading by a native English speaker is necessary.

We have checked the spelling of the manuscript with the help of a native English speaker.

We have modified the figure1, rather than adding another one, increasing details.

We have better detailed the aim of the review(The Objectives)

L21, should read “has been poorly studied.”

We have modified in “has been poorly studied”

L30, “such as”

We have modified adding “such as”

L30, the last statement needs references.

We have added the appropriate reference.

L33, The sentence “For this reason...” needs to be rewritten.

We rewrote the sentence:“Consequently, it is thought that a decreased quantity of oral nitrate reducing bacteria and an increased quantity of pathogenic bacteria are responsible for a correlation between oral hygiene and chronic periodontitis and, at a later stage, cardiovascular diseases (CVD).”

L47, The sentence “The pathway...” needs to be rewritten.

We rewrote the sentence:“This pathway is capable to reduce BP in both healthy young and old adults".

L49, should read “is responsible for”

We have modified in “is responsible for”

We have deleted the sentence“Instead the association between reducing bacteria and hypertension has never been studied so far.”

L53, is it BP or hypertension status? Which one?

We rewrote the sentence:“The aim of this review is to investigate the correlation between the oral nitrate-reducing bacteria and measured BP in healthy and hypertensive individuals.”

L58, “identified” is not a proper word.

We have changed in “recognized”

L61, check spelling.

We have checked spelling

L62, “more or less” should not appear in a scientific article.

We have changed in“somewhat”

L64, Sentence “Instead...” needs to be rewritten.

We rewrote the sentence:“On the contrary, deoxygenated hemoglobin and myoglobin oxidize NO and nitrite to nitrate”

L68, Sentence “ nNOS...[6]” needs to be rewritten.

We rewrote the sentence:“nNOS was discovered in the brain and is involved in central and peripheral neuronal signal [18]. iNOS, the inducible form of NOS, detected in macrophages, produces high levels of NO when such cells are activated. These high levels of NO can become toxic towards bacteria, virus and tumour cells.”

L81, “NOS...(GC)” needs to be rewritten.

We rewrote the sentence:“NOS forms S-nitrosothiols; NO reduced from S-nitrosothiols and NO2-can exert classical cytotoxic and cyclic GMP (cGMP)-dependent effects, such as vascular smooth muscle relaxation.”

2. It does not appear that the authors understand the fundamental chemistry: even the essential chemical formulasare all written wrong!

In NO2 and NO3, 2 and 3 should be subscripted.

NO3-if the negative sign is for negative charge, it needs to be superscripted.

Fig. 1 same problem: NO-3, 3 needs to be put right after O, NO-2, 2 needs to be put right after O.

We have modified the chemical terms/formulas in both text and figure.

2. The manuscript is mere literature report, with minimal effort of dissemination of useful information into strong arguments supporting the objectives the authors have set out to do, demonstrating “How periodontal disease and presence of nitric oxide reducing oral bacteria can affect blood pressure.” Hence, the correlation is not immediately available and the Conclusion remains unsupported.And with statements, such as “Instead the association between reducing bacteria and hypertension has never been studied so far,” the authors failed to present a strong case.

Unfortunately, the few published studies that support the Conclusion, were given very little account. More details should be given.

We have modified the Conclusion according to Reviewer’s suggestion.

We enriched the conclusions with more evidence about correlation between nitric oxide reducing oral bacteria and blood pressure,underlining how the use of antiseptic mouthwash can affect blood pressure.We also deleted the sentence“Instead the association between reducing bacteria and hypertension has never been studied so far.”

Round 2

Reviewer 2 Report

Although the authors have modified the sentences this reviewer had pointed out after careful word-by-word reading of ONLY THE FIRST TWO PAGES, and claimed “We have checked the spelling of the manuscript with the help of a native English speaker,” many sentences still have grammatical errors (below are a few examples amongst the many). Perhaps the authors can trouble your “native English speaker” colleague again to go through the whole manuscript and check the English (grammar usage and style) thoroughly and then acknowledge their proofreading in proper ACKNOWLEDGEMENT section, and simply leave the spelling check job to a word processor, e.g. Microsoft word does a decent job.

L23, “has poorly been studied.”

L305, “associated to..”

L321, “Nitrate- and nitrite-reducing...”

L330, “Remains to be…”

There are more. Grammatically wrong sentences can not convey the correct messages, let alone scientific knowledge.

Author Response

To the Reviewer #2

We thank you for reviewing the manuscript and making comments and suggestions.

We have checked the spelling of the manuscript, and we have modified some sentences.

We hope that we have satisfied the reviewer's suggestions.

Sincerely,

Maria Cristina Curia, PhD